# Advances in Immunotherapy for the Treatment of Adult Glioblastoma: Overcoming Chemical and Physical Barriers

**DOI:** 10.3390/cancers14071627

**Published:** 2022-03-23

**Authors:** Mirna Lechpammer, Rohan Rao, Sanjit Shah, Mona Mirheydari, Debanjan Bhattacharya, Abigail Koehler, Donatien Kamdem Toukam, Kevin J. Haworth, Daniel Pomeranz Krummel, Soma Sengupta

**Affiliations:** 1Foundation Medicine, Inc., Cambridge, MA 02141, USA; mlechpammer@foundationmedicine.com; 2Department of Biochemistry and Molecular Pharmacology, New York University Grossman School of Medicine, New York, NY 10016, USA; 3Department of Neurology and Rehabilitation Medicine, University of Cincinnati College of Medicine, Cincinnati, OH 45267, USA; raorn@mail.uc.edu (R.R.); bhattadj@ucmail.uc.edu (D.B.); koehleai@ucmail.uc.edu (A.K.); kamdemde@ucmail.uc.edu (D.K.T.); 4Department of Neurosurgery, University of Cincinnati College of Medicine, Cincinnati, OH 45267, USA; shahs6@ucmail.uc.edu; 5Department of Internal Medicine, Division of Cardiovascular Health and Disease, University of Cincinnati College of Medicine, Cincinnati, OH 45267, USA; mirheyma@ucmail.uc.edu (M.M.); hawortkn@ucmail.uc.edu (K.J.H.)

**Keywords:** gliomas, brain tumors, immunotherapy, immune checkpoint inhibitors, ultrasound

## Abstract

**Simple Summary:**

The poor prognosis for glioblastoma (GBM) despite the existence of a standard-of-care treatment of resection, radiotherapy, and adjuvant chemotherapy has necessitated the exploration of other therapeutic avenues. One particularly promising avenue is an immunotherapeutic approach in which the body′s immune system is artificially stimulated to directly identify and attack the tumor cells. A variety of methods including immune checkpoint inhibition, T-cell transfer, vaccination, and a viral approach are being developed for GBM. Barriers such as tumor heterogeneity, the physical blood–brain barrier, the immunosuppressive nature of GBM, and the limited number of identifiable GBM-specific targets have reduced the efficacy of the aforementioned approaches. In the following review, we document the advances in immunotherapy, the barriers to implementation, and the development of a new technology (microbubble-enhanced focused ultrasound) to overcome the physical barriers to immunotherapy.

**Abstract:**

Glioblastoma, or glioblastoma multiforme (GBM, WHO Grade IV), is a highly aggressive adult glioma. Despite extensive efforts to improve treatment, the current standard-of-care (SOC) regimen, which consists of maximal resection, radiotherapy, and temozolomide (TMZ), achieves only a 12–15 month survival. The clinical improvements achieved through immunotherapy in several extracranial solid tumors, including non-small-cell lung cancer, melanoma, and non-Hodgkin lymphoma, inspired investigations to pursue various immunotherapeutic interventions in adult glioblastoma patients. Despite some encouraging reports from preclinical and early-stage clinical trials, none of the tested agents have been convincing in Phase III clinical trials. One, but not the only, factor that is accountable for the slow progress is the blood–brain barrier, which prevents most antitumor drugs from reaching the target in appreciable amounts. Herein, we review the current state of immunotherapy in glioblastoma and discuss the significant challenges that prevent advancement. We also provide thoughts on steps that may be taken to remediate these challenges, including the application of ultrasound technologies.

## 1. Introduction

Glioblastoma is one of the most common primary malignant adult brain tumors, typified by its aggressiveness. The current standard-of-care treatment includes maximal resection and radiotherapy, followed by adjuvant chemotherapy with the DNA alkylator temozolomide [1]. The median overall survival (MOS) following GBM diagnosis is 12–15 months [1]. A multitude of factors complicates the treatment of GBM including (1) the heterogeneous nature of the tumors, both within a patient and between patients; and (2) the highly impermeable blood–brain barrier (BBB), which limits the effective delivery of many standard therapeutics.

A recent promising advancement has been an immunotherapeutic approach, which may involve either antagonizing the tumor′s inherent immune-suppressive properties or, conversely, inducing a glioma-specific immune response using either exogenous or endogenous agents. Immunotherapy has recently been popularized by the impressive outcomes in hematogenous malignancies [2]. However, for immunotherapy to be successful in solid tumors such as GBM, it must overcome tumor heterogeneity and the physical barriers imposed by the BBB and the tumor microenvironment (TME).

To overcome heterogeneity, key mutations which underlie GBM pathogenesis are continuously being elucidated so that targeted immunotherapeutics can be more effectively developed to combat the complexity of GBM. To overcome the BBB, ultrasound is being developed as a modality for transiently and noninvasively disrupting the BBB for the passage of therapeutics [3,4,5,6]. In the following review, we detail the advances in immunotherapies and how their efficacy can be enhanced by ultrasound technologies.

## 2. Current Immunotherapy Options and Developments

The human immune system is a complex regulatory environment that must constantly be able to distinguish between “self” and foreign matter. The immune system can be split into “innate” and “adaptive” immunity. Innate immunity does not improve with repeated encounters and consists of phagocytic cells (neutrophils, monocytes) and pro-inflammatory cells (eosinophils, basophils, and mast cells) [7]. Adaptive immunity learns and improves upon repeated exposure to pathogens. The main players in adaptive immunity are B and T lymphocytes, which produce antigen-specific immunoglobulins and induce foreign cell lysis [7]. Following activation, part of the immune system′s natural response is to return the hyperactive immune response to basal levels. Cells such as regulatory T-cells (Tregs) release anti-inflammatory cytokines leading to a diminished immune response [8]. Similarly, cell–cell signaling via inhibitory immunoreceptors such as PD-1, CTLA-4, LAG3, TIM3, TIGIT, and BTLA can attenuate an upregulated immune response [9]. The most promising immunotherapy approaches to treating glioblastoma are immune checkpoint inhibition [10,11], T-cell transfer therapy [12], vaccination [13], and oncolytic virus therapy (OVT). These methods harness the immune system to recognize and focally target tumor cells.

### 2.1. Immune Checkpoint Inhibitors

Immune checkpoint inhibitors (ICIs) avert the inactivation of CD8+ T-cells by preventing checkpoint receptors from binding with their ligands (Figure 1A). The critical immune checkpoint targets for ICIs in glioma include programmed cell death protein-1 (PD-1), programmed cell death ligand-1 (PD-L1), and cytotoxic T-lymphocyte-associated antigen 4 (CTLA-4). In the canonical pathway, when the PD-L1 ligand on the target cells interacts with the PD-1 receptor on the T-cells, intracellular tyrosine residues on the PD-1 cytoplasmic region lead to recruitment of Src homology 2 domain-containing protein tyrosine phosphatase-2 (SHP-2) [14]. This causes spleen tyrosine kinase (Syk) and phospholipid inositol-3-kinase (PI3K) to be phosphorylated, resulting in T-cell exhaustion and a suppressed immune response [14,15]. Glioblastoma cells can co-opt this machinery by overexpressing PD-L1, thereby evading the immune response [16]. Through a similar mechanism, glioblastoma cells also can upregulate CTLA-4, which promotes T-cell anergy through blockade of the B7/CD28 co-stimulatory signal [17].

A Phase III trial (CheckMate 143) compared the efficacy of the PD-1 inhibitor nivolumab, either alone or in combination with the CTLA-4 inhibitor ipilimumab, versus the vascular endothelial growth factor (VEGF) inhibitor bevacizumab in a subset of patients with recurrent glioblastoma (Table 1) [18]. Nivolumab was not superior to bevacizumab; the MOS was 9.8 and 10.0 months under nivolumab and bevacizumab, respectively. In another Phase III trial (CheckMate 498), nivolumab combined with radiotherapy also failed to prolong survival in patients with newly diagnosed MGMT-unmethylated glioblastoma compared with the SOC [19]. However, in a small randomized Phase II trial with 35 recurrent glioblastoma patients, the administration of the PD-1 inhibitor pembrolizumab before and after surgery significantly prolonged overall survival compared with adjuvant administration alone: 13.7 vs. 7.5 months, respectively [20].

In a single-arm Phase II study, a regimen consisting of avelumab (a monoclonal PD-L1 antibody) and axitinib (a tyrosine kinase inhibitor targeted against multiple VEGF receptors) was prescribed to patients with recurrent glioblastoma. It was well tolerated but failed to meet the study threshold for activity [21]. In another Phase II study, the addition of the PD-L1 inhibitor durvalumab to standard therapy moderately prolonged survival in patients with newly diagnosed MGMT-unmethylated glioblastoma compared with historical controls: 15.1 vs. 12.7 months, respectively [22]. The ambiguity of these clinical trial results involving CTLA-4 or PD-L1 inhibitors suggests the need for further research into a combinatorial approach, which may be feasible, given that each pathway leads to unique alterations in cytokine release [34].

### 2.2. T-Cell Transfer Therapies

T-cell transfer therapy or adoptive T-cell therapy is a type of immunotherapy that encompasses two main approaches: tumor-infiltrating lymphocyte (TIL) therapy and chimeric antigen receptor (CAR) T-cell therapy (Figure 1B).

In TIL therapy, T-lymphocytes invading the TME are collected via routine biopsy or surgery, isolated using fluorescence-activated cell sorting (FACS), and then selectively expanded using IL-2 stimulation [35,36,37]. The logic behind this approach is that T-cells found in or near the tumor already have a “proven track record” for identifying cancerous cells, but there are too few of them to overcome immunosuppression. Moreover, TIL therapy significantly reduces off-target effects due to their inherent specificity to the tumor [37]. Mathewson et al. performed single-cell transcriptome sequencing in a group of patients with isocitrate dehydrogenase (IDH)-mutant and IDH-wildtype glioblastoma [38]. They described the potential effectors of anti-tumor immunity in a population of cytotoxic TILs expressing several natural killer (NK) cell genes, including the CD161-encoding gene KLRB1. The inactivation of KLRB1 or antibody-mediated CD161 blockade resulted in increased T-cell cytotoxicity against tumor cells in vitro and an enhanced response in vivo.

CAR T-cell therapy introduces synthetic T-cell receptors into T-cells, which confer the ability to recognize tumor-specific surface antigens and initiate an MHC-independent immune response [39,40,41]. CAR T-cell therapy has had great efficacy in hematogenous malignancies but has been difficult to implement in solid tumors due to the immunosuppressive environment of the TME [2,40]. Moreover, solid tumors lack highly specific surface antigens, which can lead to numerous off-target effects when using CAR T-cell therapy. Two small Phase I trials tested CAR T-cell therapy in EGFRvIII-positive recurrent glioblastoma. Although EGFRvIII-targeted CAR T-cells found their way from peripheral blood to the tumor, no meaningful response was detected [23,42]. This lack of response to anti-EGFRvIII CAR T-cells may be attributable to the significant intra- and inter-tumoral heterogeneity of EGFRvIII expression in glioblastoma as well as to adaptive changes in the local TME, which include changes in antigen expression over time. For instance, following treatment, EGFRvIII was lost in a group of patients.

### 2.3. Vaccination

Tumor vaccines elicit an immune response against one or several tumor antigens (Figure 1C). Vaccines usually consist of peptides or proteins, but may also constitute antigen-laden dendritic cells. Immunostimulants such as poly ICLC are often co-administered with tumor vaccines to enhance adaptive immunity.

In a single-arm, multicenter, open-label Phase I trial performed in patients with newly diagnosed Grade 3 and 4 IDH1-mutant astrocytoma, an IDH1-specific peptide vaccine induced an immune response in 30 out of 32 (93.3%) patients [24]. The 3-year progression-free and overall survival rates were 63% and 84%, respectively. The 2-year progression-free rate among patients with an immune response was 82%, while the two patients without an immune response had tumor progression within 2 years of diagnosis.

Another vaccine approach involves a vaccination against survivin, an antiapoptotic protein expressed by many tumor types [25,43,44]. Survivin expression in GBM has been associated with increased recurrence, chemotherapy resistance, and poor overall prognosis [25,43,44,45,46]. The SurVaxM vaccine contains a synthetic long peptide mimic that spans the human survivin protein sequence; it expresses MHC Class I epitopes and stimulates the MHC Class II-restricted T-cell responses required for cytotoxic CD8+ T-cell activity against tumors [25]. A Phase I trial of SurVaxM against recurrent GBM demonstrated no serious adverse events and prolonged overall survival following vaccination (86.6 weeks) compared with historical overall survival (30 weeks) [25]. A subsequent study identified that glioma patients routinely expressed elevated serum levels of CD9+/GFAP+/SVN+ exosomes, associated with tumor progression, compared with healthy controls [47]. Patients treated with antisurvivin therapy showed decreased levels of these exosomes. Monitoring of CD9+/GFAP+/SVN+ exosomes may be a promising adjunct to the use of MRI in disease surveillance. Current trials are underway to evaluate SurVaxM’s efficacy in newly diagnosed GBM [26]. However, identifying plausible new vaccine targets for GBM remains difficult due to the heterogeneity of GBM tumors.

### 2.4. Oncolytic Virus Therapy

In recent years, the use of OVT has shown promise in the treatment of GBMs. OVT utilizes intratumoral delivery of viral vectors to either deliver oncolytic gene therapy into the TME or to cause direct cytotoxicity through viral infection and replication [48,49]. OVT also has pro-immunogenic effects due to the induction of immunogenic cell death (ICD) in infected tumor cells. In ICD, the destruction of tumor cells by OVT leads to the release of antigenic molecules into the TME which both recruits and activates local dendritic cells, with the subsequent stimulation of specific T-cells [49].

The earliest trials of oncolytic therapy in GBM used murine fibroblasts to deliver the replication-defective herpes simplex virus 1 (HSV1) thymidine kinase (tk) gene to GBMs, which conferred increased chemosensitivity to antiviral agents such as acyclovir, ganciclovir, and valganciclovir [48,50]. However, this trial failed to show prolonged survival in the OVT group, which was hypothesized to be the result of low gene transduction rates due to the nonmigratory nature of murine fibroblasts [50]. More recently, a genetically engineered replication selective HSV1 virus, G207, has shown safety and efficacy in clinical trials. G207 contains a deletion of the diploid γ_1_34.5 neurovirulence gene and has viral ribonucleotide reductase (UL 39) disabled by the insertion of *Escherichia coli* lacZ. This allows for conditional replication in tumor cells while preventing the infection of normal cells [51]. A Phase I trial showed a median survival of 15.9 months in 13 GBM patients treated with intratumoral G207, with no evidence of HSV encephalitis [27,52]. A separate Phase I trial demonstrated the safety of G207 administration in conjunction with radiotherapy [27], while a more recent trial showed its safety in the treatment of pediatric high-grade gliomas [28]. HSV-vector mediated delivery of gene therapy offers significant promise in the treatment of GBM, and a current Phase I trial is investigating the use of a new drug, rQnestin34.5v.2, after a preclinical study suggested its low toxicity to humans [53,54].

Another development in OVT was the use of intratumoral injection of aglatimagene besadenovec (GliatakTM), a replication-defective adenovirus vector-mediated delivery of HSV1-tk (AdV-tk), in conjunction with subsequent valaciclovir therapy. Phase I trials of Gliatak conducted by Chiocca and colleagues demonstrated the safety of the therapy and an impressive radiographic response [55], while the Phase II trial showed a statistically significant improvement in the MOS of GBM patients treated with Gliatak after gross total resection (GTR) compared with patients treated with the standard of care after gross total resection (25.1 months vs. 16.3 months, respectively) [29]. Importantly, the survival benefit was even further improved at 2 and 3 years compared with the standard of care treatment, but no difference was noted if the resection was subtotal [29]. However, another Phase III clinical trial named the Aspect trial, which utilized AdV-tk, showed no significant improvement in overall survival when patients were treated with intratumoral injections of AdV-tk compared with the standard of care treatment group [30]. It should be noted that the ASPECT trial had uneven use of temozolomide, and radiotherapy was not administered concomitantly with the gene therapy [30]. Yet another Phase I trial evaluated the use of a human interferon-β-expressing adenovirus vector (Ad.hIFN-β). Intratumoral injection of Ad.hIFN-β was associated with a dose-related induction of apoptosis within tumors, but several patients experienced adverse effects and one patient experienced two serious dose-related adverse effects [56]. Ultimately, further investigation into adenovirus vectors is required.

The use of a live attenuated form of poliovirus has recently been studied as well. A Phase II clinical trial demonstrated that PVSRIPO, a live attenuated poliovirus Type 1 vaccine with its cognate internal ribosome entry site replaced by that of human rhinovirus Type 2 conferred an overall survival benefit [31]. Specifically, this randomized controlled trial (RCT) showed that the group treated with PVSRIPO had an overall survival rate of 21% at both 24 and 36 months, compared with 14% and 4% in the control group, respectively [31]. The foreign ribosomal entry site on PVSRIPO causes neuronal incompetence and ablates neurovirulence [57]. The effects of PVSRIPO are mediated by CD155, a Type 1 transmembrane glycoprotein receptor that is more commonly known as the poliovirus receptor [31,58,59,60]. CD155 is almost ubiquitously upregulated in solid tumors, including GBM, and it regulates natural killer (NK) cells and is part of the Ig-superfamily adhesion family response for cell motility and invasiveness [58,60,61]. When the PV capsid binds to CD155, the capsid protein is extruded and ultimately initiates the transfer of the viral RNA genome to the cytoplasm, then subsequently allows for the translation of the RNA and mediates the viral oncolytic effects [62]. Additional Phase II studies for PVSRIPO in conjunction with additional drugs are underway, with Phase III studies likely to commence in the foreseeable future.

Translating the success of early Phase I and Phase II trials to widespread clinical use has been challenging. Phase I and Phase II trials of the drug Toca 511 (Vocimagene amiretrorepvec), a γ retroviral replicating vector encoding a transgene for an optimized yeast cytosine deaminase, demonstrated both early safety and efficacy, with prolonged overall survival and complete responses in recurrent high-grade glioma and GBM compared with accepted survival rates in the literature [32]. However, in the Phase III arm of the clinical trial, the overall survival for patients treated with Toca 511 was 11 months compared with 12 months in the patient group receiving the standard-of-care treatment, with no significant difference between the two groups [33]. Toca 511′s Phase III failure underscores how challenging the introduction of new GBM therapies into the market has been. Several obstacles underlie these challenges in translating OVT into widespread clinical use. Pre-existing antibodies and the circulating complement in the peripheral vasculature may neutralize OVT particles before they are successfully delivered into the TME [63]. Moreover, uptake into nontarget organs (e.g., the liver) is a common barrier to efficient delivery [63]. As with the other therapeutic modalities, the BBB is a major obstacle to the effective delivery of any exogenous therapeutics.

## 3. Challenges to Immunotherapy

The equivocal results of clinical studies testing the four aforementioned immunotherapeutic agents in glioblastoma, compared with other solid tumor types, are largely due to three key factors of immune resistance: the blood–brain barrier and the brain–tumor barrier (BTB), the immunosuppressive microenvironment, and the low tumor mutational burden (TMB) of glioblastoma.

### 3.1. Blood–Brain/Brain–Tumor Barriers

The BBB is a semipermeable physiologic border that isolates the blood from the cerebrospinal fluid and the internal environment of the central nervous system (CNS) to preserve homeostasis and maintain normal brain function. The BBB comprises endothelial cells of the capillary wall, astrocyte endfeet wrapping the capillary, and pericytes of the capillary basement membrane (Figure 2) [64,65]. The endothelial cells making up the BBB are tightly linked through a series of tight junctions which prevent the paracellular passage of most large molecules. Astrocytes are a glial population that is well-known for regulating the synaptic junction but play a diverse set of roles in the CNS, one of which is regulation of the BBB. The insulation provided by astrocytic endfeet has shown to be compromised in various neural proteinopathies (e.g., Parkinson′s disease), where the lack of integrity of the BBB can lead to the accumulation of pathogenic solutes [66].

A subpopulation of GBM stem cells is localized in proximity to microvascular capillaries and subvert cerebrovascular tissue function to turn the BBB into a BTB. Even after a growing tumor damages the BBB, the newly formed BTB prohibits the optimal accumulation of drugs in the tumor (Figure 2). The BBB–BTB acts to shield the TME from therapeutics [67,68].

### 3.2. The Immune-Suppressive Microenvironment

The glioblastoma microenvironment is dominated by immunosuppressive tumor-associated macrophages (TAMs), which eliminate the effect of immunotherapy and promote tumor growth. Colony-stimulating factor 1 (CSF-1) and chemokine (C-C motif) ligand 2 (CCL2), which are overexpressed in glioblastoma, attract macrophages and determine their behavior. TAMs suppress antitumor immunity via at least two mechanisms: (1) the production of arginase and inducible nitric oxide synthase (iNOS); (2) surface expression of IL-4Rα. Arginase and iNOS restrict the proliferation of T-cells by depleting essential amino acids from the extracellular space [69]. Activation of IL-4Rα leads to the overexpression of transforming growth factor (TGF)-β, which, in turn, suppresses the IL-2-dependent survival of CD8+ T-cells and diminishes their activity by curbing the production of several effectors and immune-stimulatory molecules, including granzymes A and B, perforin, IL-6, IL-10, and IFN-γ [70]. Furthermore, TGF-β promotes the differentiation of naïve T-cells into (suppressor) Tregs. In GBM patients, the adenosine receptor pathway (A2aR/CD39/CD73), followed by PD-1, was found to be the most frequent immunomodulatory target in CD8+ cytotoxic T-cells obtained from the TME. Among various other immune markers profiled in GBM patients, A2aR expression was higher in TILs compared with the peripheral blood mononuclear cells (PBMCs) of GBM patients and PBMCs obtained from healthy donors [71]. The GBM TME induces hypoxia and cellular stress, leading to increased production of ATP, followed by its conversion to AMP by ectonucleoside CD39, ultimately resulting in the production of adenosine [72]. Increased levels of adenosine in the GBM TME suppress the effector function of TILs and recruit TAMs, contributing to immunosuppression [71,72].

Typically, leukocytes are absent from the brain parenchyma. However, a small number of T-cells can be found in the cerebrospinal fluid, choroid plexus stroma, and subarachnoid and perivascular spaces. Some of these T-cells escape from the capillaries in the event of a primary malignant brain tumor. Higher numbers of intratumoral CD8+ T-cells have been shown to correlate with better prognosis in several cancer types, including glioblastoma [73,74,75]. However, in glioblastoma, CD8+ T-cells comprise only 0–12% of all cells in the tumor. Besides, a significant fraction of these CD8+ T-cells show signs of exhaustion [76,77].

Cancer stem cells (CSCs) are a cellular subpopulation of GBMs which exhibit a unique form of immunosuppression. These CSCs are capable of self-renewal and differentiation, thereby repopulating the tumor niche [78]. They serve as a particularly complex barrier to treatment, as they are difficult to completely resect and display chemo/radioresistance [79]. CSCs can be isolated through the expression of the surface markers CD24, CD34, CD44, CD47, CD90, and CD133 [80]. CSCs have been shown to alter the immune microenvironment through the recruitment of TAMs, which help to maintain the self-renewal and maintenance capabilities of CSCs [81,82]. Additionally, CSCs display the ability to inhibit the proliferation of TILs through the upregulation of PD-L1 [78,81]. Conversely, CSCs may serve as a viable vaccine target, given that CSC lysates are more effective at generating a dendritic cell vaccine compared with whole tumor cell lysates [81,83]. The double-edged nature of CSCs warrants further investigation in regards to immunotherapy.

Further complicating the inherent GBM immunosuppressive environment, patient-dependent lifestyle choices can impact the immune system′s ability to mount a successful immune response. For example, in obesity, the immune system cellular profile changes from an anti-inflammatory/regulatory to a pro-inflammatory profile [84]. Interestingly, the data has been equivocal as to whether this shift to a pro-inflammatory state in obesity is beneficial, with studies both showing increased survival in obese patients with melanoma treated with ICIs, and obesity leading to tumor progression through T-cell aging [85,86,87]. Another common confounding lifestyle choice is the patient’s smoking history. Smoking both increases the TMB of tumors and alters the immunogenic microenvironment in a site-dependent manner [85,88,89,90]. Patient lifestyle choices must be contextualized in both the specific tumor subtype and anatomical localization.

### 3.3. Low Tumor Mutational Burden

TMB, defined as the total number of nonsynonymous mutations per coding area of a tumor′s genome, is a promising predictor of the response to treatment with immune checkpoint inhibitors in various cancers, including melanoma, renal cell carcinoma, and non-small-cell lung cancer [91,92]. Initially, TMB was determined using whole-exome sequencing of tumor samples, with targeted panel sequencing being currently explored [93,94]. The neoantigen and antigen burden in glioblastoma is generally low. Glioblastoma harbors a relatively insignificant number of mutations compared with immunogenic tumors, such as non-small cell lung cancer or melanoma. Only a few mutation-derived neoantigens have been predicted in glioma [13,95,96,97,98]. The expression levels of other, non-mutated targets (e.g., cancer germline antigens) are usually low. Recently, a study using multi-omics data from The Cancer Genome Atlas (TCGA) and the Chinese Glioma Genome Atlas (CGGA) found that TMB was an independent marker of prognosis in diffuse glioma [99].

Based on a cut-off value between 0.64 and 0.67 mutations/Mb, Wang et al. classified 654 primary glioma patients from the TCGA database into TMB-high and TMB-low groups and revealed an inverse correlation between TMB and glioma grade [99]. As expected, an analysis of the distribution of nonsynonymous mutations showed that the TMB-high group had a higher incidence of mutations typical for glioblastoma (PTEN: 29% vs. 5%; EGFR: 17% vs. 5%), while the opposite was true for mutations associated with low-grade gliomas (IDH1: 77% vs. 7%). The patients with elevated TMB had, on average, less favorable outcomes than the patients with decreased TMB. The MOS was 23 months in the TMB-high group. Gene set enrichment analysis in the TMB-high group revealed enrichment in transcriptional programs associated with DNA replication and the cell cycle, indicating increased proliferative activity in high-TMB gliomas, which may, in part, explain the lack of treatment effect in these tumors. Further complicating these findings is the fact that glioblastoma TMB can increase following SOC treatment [100].

Gromeier et al. performed a genomic analysis of recurrent glioblastoma biopsy samples and determined that tumors harboring low TMB were more responsive to subsequent treatment with recombinant polio virotherapy (PVSRIPO) or immune checkpoint inhibitors [101]. They found that the patients who survived longer than 20 months after PVSRIPO treatment carried a TMB of less than 0.6 mutations/Mb. Stratifying overall survival following treatment with PVSRIPO or checkpoint inhibitors based on the median TMB (1.3 mutations/Mb) verified a more favorable response in patients carrying a below-median TMB in both cohorts. The difference remained significant even after excluding patients with hypermutation (>10 mutations/Mb). Notably, a correlation between survival and TMB has not been observed in immunotherapy-naïve primary or recurrent glioblastoma.

## 4. Strategies to Enhance Immunotherapy’s Effectiveness

The failure to achieve a meaningful clinical benefit through immunotherapy exposes the flaws of the current immunotherapeutic approaches in glioblastoma. The need for strategies to increase the sensitivity of glioblastoma to immunotherapeutic agents is evident. Finding ways to increase the influx of cytotoxic T-cells to the tumor, downregulate the immunosuppressive microenvironment, and target the low immunogenicity of glioblastoma could be some of the potential next steps. In addition, reflecting on and revising disease management is warranted. Specifically, developing alternatives to steroids (such as the glucocorticoid dexamethasone) for the effective control of edema in GBM patients is potentially crucial, because this would allow us to avoid steroid-induced immunosuppression [102,103].

One of the most logical targets for improving immunotherapy’s effectiveness is the BBB, given that the BBB is known to limit immune cell infiltration and antigen presentation in glioma. Recent findings have suggested promising strategies to mitigate this limitation. A compromised BBB increases the expression of tumor-associated antigens, as evidenced by the improved responses in glioblastoma patients with both pre- and post-surgical administration of PD-L1 blockers compared with adjuvant administration alone [20].

However, although the BBB is breached in glioblastoma, the disruption is heterogeneous, and thus sufficient delivery of an intravenously administered drug, such as nivolumab, to the entire TME has not been achieved [104]. Notably, the BBB restricts the general passage of compounds heavier than 400–600 Da and those that have a charge that is not intermediate or low, significantly hampering the treatment of brain tumors and CNS diseases [105]. For reference, the molecular mass of nivolumab is 146 kDa.

Physical modalities, such as noninvasive microbubble-enhanced focused ultrasound (MB-FUS) (Figure 3), can safely and transiently alter the permeability of the BBB/BTB without directly causing changes in the tumor cells. This technology has been demonstrated preclinically in numerous species, including nonhuman primates [3,106,107,108,109,110,111,112,113,114,115], and in multiple successful Phase I and IIa clinical trials executed by several different groups [4,5,116,117,118,119,120,121,122,123]. Ultrasound-mediated BBB disruption has been observed in normal brains [3,124], brains affected by neurodegenerative diseases (e.g., Parkinson′s disease and Alzheimer′s disease) [123], and brains with tumors [4,5,6]. Together, these studies have demonstrated the robustness of the technique. The temporary increase in permeability lasts between a few hours and several days, and depends on the type and dose of the microbubbles used and the ultrasound parameters [125,126,127,128,129,130,131]. The increased permeability occurs both through the opening of the tight junctions of the endothelium and through increased transcytosis [132]. These effects are nucleated by the gentle volumetric oscillation of the microbubbles when they are exposed to low-amplitude ultrasound, with the ultrasound amplitude being within the range used for diagnostic ultrasound imaging. Care must be taken to identify the appropriate ultrasound amplitude. If the amplitudes are too low, the barrier will not be disrupted, and for amplitudes that are too high, petechial hemorrhage may occur [133]. The emissions from the oscillating microbubbles can be used to identify the appropriate amplitudes in real time, providing patient- and treatment-specific guidance and control [134,135,136,137,138,139]. Phase I clinical trials have demonstrated the safety of this technology. Oscillation of the microbubbles not only increases the permeability of the blood–brain/tumor barrier but can also establish a convective flow that enhances the delivery of chemotherapeutics [140,141,142,143].

Because of the cavitation-dependent nature of barrier disruption, specific locations of disruption in the brain can be controlled with high precision based on where the ultrasound is focused in the brain. Multiple approaches have been pursued to obtain precise ultrasound insonation. The most common approach is magnetic resonance imaging-guided focused ultrasound (MRgFUS). This approach uses a stereotactic system and concurrent MR imaging to perform a real-time guided treatment, optimizing precision. In small animal preclinical models, the focused ultrasound transducer is typically a single element and targeting is achieved by physically moving the transducer [3,144,145,146]. More advanced systems use arrays of transducer elements and ultrasound beamforming to electronically steer the beam throughout the regions of interest in the brain [147,148,149]. A key advantage of these systems is that they can account for the effects of the skull when focusing the ultrasound. Additionally, the procedure avoids mechanical brain tissue shifts and eliminates the risk of infection. This approach has been used in clinical trials with the ExAblate Neuro system from Insightec [4,5,119,123,150]. A drawback of this approach is the financial expense associated with using the MR imaging system. An alternative methodology uses neuronavigation systems for targeting focused ultrasound transducers (either single-element or multi-element arrays) [124,151,152,153]. The NaviFUS system is testing this methodology clinically [154,155]. A recent Phase I immunostimulation study demonstrated safe delivery across the BBB [6]. The study was designed to determine if the ultrasound insonation induced immunostimulation without other therapeutics being administered (e.g., a chemotherapeutic). No immunostimulation was observed in the clinical trial participants. A follow-on preclinical study determined that immunostimulation occurred if the ultrasound’s pressure amplitude was increased above those used in the clinical trial [6]. Cavitation-mediated inflammation has also been observed in other preclinical studies using neuronavigation [156]. A third clinically investigated methodology uses surgically implanted ultrasound transducers [157,158,159]. The SonoCloud systems from CarThera have demonstrated increased barrier permeability in patients with recurrent glioblastoma and a trend toward increased survival with the co-administration of carboplatin [116,117,160].

Delivery of a wide range of potential therapeutics has been demonstrated in preclinical models, including chemotherapeutics [121,161,162], adenoviruses [163,164], antibodies [165,166], nanoparticles (NPs) [142,167,168,169], and whole cells [170,171]. Guo et al. demonstrated that NPs as large as 50 nm can achieve significant extravasation into the TME with the application of focused ultrasound [142]. NPs have a wide variety of formulations. Guo et al. used them as a lipid-based encapsulation method to protect therapeutic payloads from degradation as they traversed the vasculature to the TME. Their study also demonstrated that focused ultrasound delivery of RNA-loaded NPs significantly downregulated the expression of an oncogenic mRNA [169,172]. NPs have a use in immunotherapy, as they can be combined with anti-PD-L1 antibodies to focally target drug delivery to the TME [173,174,175]. Similarly, groups have used NPs to deliver CAR-T-cells in a mouse model of glioma [174,176]. NPs could also have a use in the delivery of vaccines or OVT, given the previously discussed barriers to the effective delivery of these therapies. Ultrasound-mediated delivery to specifically induce immune modulation and therapy has been previously described [146]. Approaches include the passage of IL-12 [177], immune checkpoint inhibitors [106,116,178,179,180,181], and natural killer cells [182]. In addition to transient disruption facilitating the diffusion of therapeutics into the brain, disruption of the BBB can also enable the release of tumor biomarkers, which can assist in assessing the treatment response [183].

A concern of ultrasound-mediated therapy is the potential adverse effects of multiple sonications. Park et al. performed repeated MRgFUS on patients with GBM receiving TMZ and found no clinical adverse effects during six ultrasound insonations [117]. Another concern is the risk of RBC extravasation due to permeabilization of the brain’s vasculature. However, fine control of the FUS parameters can avoid this risk [184,185]. Finally, FUS-induced mild inflammatory responses have been reported, with some variability in the literature [111,186]. In fact, some of these groups reported an increase in IL-12-mediated immune recognition, which could enhance an immunotherapeutic approach [177,187].

While there has been a significant and deserved emphasis on focused ultrasound to transiently permeabilize the blood–brain barrier, ablative ultrasound therapies can also enhance immune checkpoint inhibition [188]. Thermal ablative ultrasound therapy uses high-intensity focused ultrasound to increase the local temperature to 60 °C or higher to induce coagulative necrosis. It has been safely used in the brain to ablate neuronal tracks underlying the pathogenesis of essential tremor [189,190,191] and also to treat chronic neuropathic pain [192]. Preclinical evidence has indicated that ultrasound thermal ablation may work adjunctively with immune checkpoint inhibitors [193,194]. Furthermore, mechanically ablative ultrasound therapy (histotripsy) can also potentially enhance immune checkpoint inhibitors by stimulating nonimmunogenic “cold” tumors into becoming “hot” immunogenic tumors [195,196]. Common obstacles to this treatment approach are the interference of uniform ultrasound wave propagation through bone and gas, and organ movement during treatment, leading to collateral tissue damage [197].

## 5. Conclusions and Future Directions

Immunotherapy has recently become a highly researched potential therapeutic avenue for glioblastoma. The four main approaches are immune checkpoint inhibition, T-cell transfer therapy, vaccination, and oncolytic viral therapy. In regards to immune checkpoint inhibition, CTLA-4 and PD-L1 inhibitors have entered clinical trials, but inconsistent effects on prolonging the median survival time have slowed progress in this area of immunotherapy. T-cell transfer therapy has similarly drawn interest due to its success in hematogenous tumors. However, in solid cell tumors, clinical trials have shown equivocal results, likely due to a combination of low T-cell penetration of the TME and the need for a more diverse array of GBM molecular targets. Research is already underway for a bi- or tri-CAR T-cell approach in which a single T-cell can have multiple antigenic targets. Groups have started developing trivalent CAR T-cells simultaneously targeting HER2, IL-13Rα2, and EphA2 in murine models of GBM [198]. A vaccine approach has gained public attention due to the recent success of the SARS-CoV-2 mRNA vaccine. The SurVaxM vaccine against the oncoprotein survivin has had moderate success in recurrent GBM and warrants further exploration of other vaccine targets. Finally, several OVT therapies have entered clinical trials, with some exciting successes due to OVT′s unique ability to both induce tumor lysis and promote an immunogenic response. The main concern with OVT is the potential for normal cell infection.

Thus far, the clinical improvements achieved with the aforementioned immunotherapies for treating several extracranial solid tumors have been modest. The lack of clinical improvements can be attributed to several factors, including the low TMB of GBM, the immunosuppressive features of GBM, tumor heterogeneity [199], and the BBB and BTB. Critical advances need to be made in finding GBM-specific antigens for targeted immunotherapy, which may suggest the need for a combinatorial approach. Moreover, given the heterogeneity of GBM, specific subsets of GBM patients may preferentially benefit from certain immunotherapies; the challenge lies in determining which subpopulations would best benefit from which immunotherapies. Furthermore, advances in noninvasive MB-FUS may provide the transient permeabilization necessary to deliver these therapeutics while bypassing the BBB and directly targeting the TME. Advances have been made in NPs delivering mRNA payloads via a technique termed “selective organ targeting” (SORT) [200]. SORT molecules are added to the lipid nanoparticles’ outer layer, which aids in their delivery to specific organs. These targeting mechanisms would allow for better delivery of therapeutics to the TME. While advances need to be made before widespread clinical success can be achieved, immunotherapy combined with ultrasound-mediated delivery serves as a highly promising avenue for treating GBM and ultimately improving patient outcomes.

## Figures and Tables

**Figure 1 cancers-14-01627-f001:**
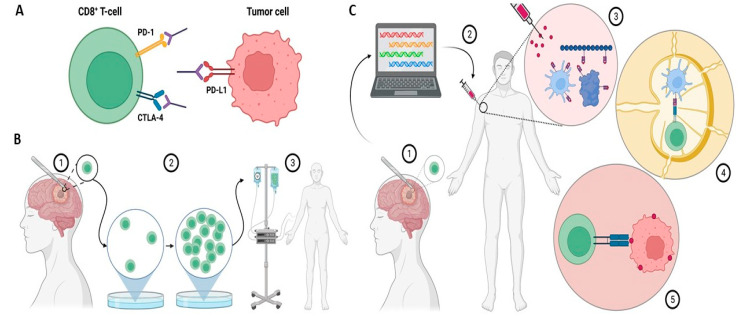
(**A**) Immune checkpoint inhibitors bind to and inhibit immunosuppressive molecules on either T-cells or tumor cells. This dampens tumor cells′ ability to evade the immune system. (**B**) (1) In tumor-infiltrating lymphocyte (TIL) therapy, T-cells from the tumor microenvironment are isolated following surgical resection. (2) Isolated T-cells are clonally expanded by using IL-2 stimulation. (3) Expanded T-cells are reintroduced to the patient. (**C**) (1) In the vaccine approach, a resected tumor biopsy is taken from the patient and sequenced to identify neoantigens. (2) Neoantigens are then delivered via a vaccine. (3) At the site of injection, neoantigens stimulate antigen-presenting cells (APCs). (4) In the lymph node, APCs present T-cells with neoantigens. (5) Activated T-cells attack cancer cells. Created with BioRender.com.

**Figure 2 cancers-14-01627-f002:**
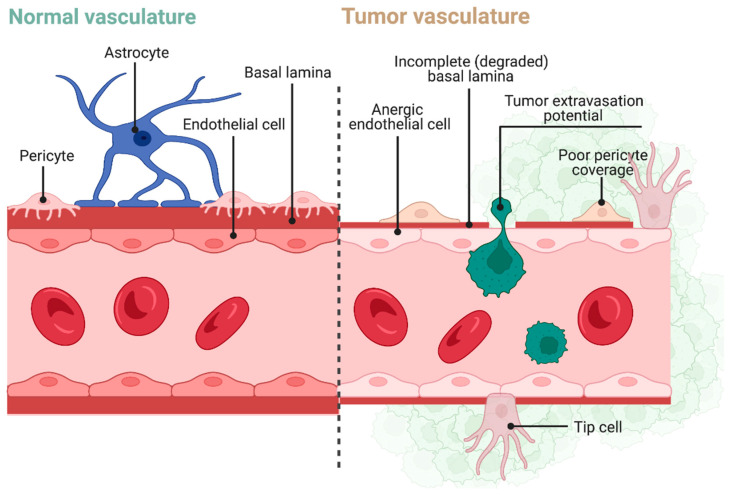
The left panel shows the normal anatomy of the blood–brain barrier in which tight junctions exist between endothelial cells to prevent the passage of most therapeutics into the brain parenchyma. This basic structure is supported by astrocytes and pericytes, which help maintain and regulate these tight junctions. The right panel shows the pathology of the BBB induced by tumor growth. For one, there is increased permeability of the endothelial cells’ tight junctions, permitting tumor cell extravasation. There is an atrophied basal lamina, which contributes to anergic endothelial cells. Finally, pericytes are both fewer and display an abnormal morphology. The combination of these factors can promote tumor migration and growth. Created with BioRender.com.

**Figure 3 cancers-14-01627-f003:**
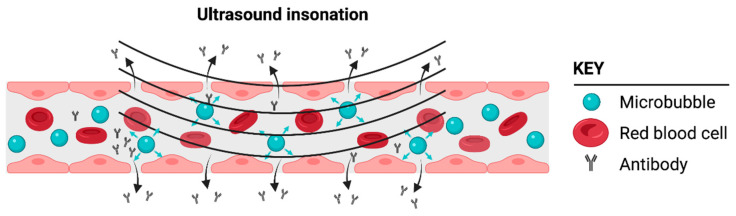
Cartoon illustrating how microbubbles can induce a focal disruption or opening of the blood–brain barrier (BBB), thus enabling the delivery of a biologic such as a monoclonal antibody. Microbubbles flow through the normal vasculature or vasculature supplying the glioblastoma tumor microenvironment (TME). Only the microbubbles in the vasculature exposed to ultrasound insonation enable BBB/BTB disruption following ultrasound insonation. Created with BioRender.com.

**Table 1 cancers-14-01627-t001:** Overview of current and in-progress immunotherapeutic clinical trials.

Trial Number	Study Design	Trial Details	Patient Number	Reference
NCT02017717	Phase III R, PA	Comparison of the PD-1 inhibitor nivolumab, with and without the CTLA-4 inhibitor ipilimumab, versus the VEGF inhibitor bevacizumab. The primary outcome is overall survival. Study is in progress.	530	[18]
NCT03291314	Phase II PA	A study of the combination of the anti-PD-L1 molecule avelumab and the VEGF inhibitor axitinib on the progression of GBM. The 6-month progression-free survival (PFS) was 18% (95% CI 4–33, *n* = 27), which did not meet the threshold for justifying further investigation.	52	[21]
NCT02336165	Phase II PA	A study of the anti-PD-L1 molecule durvalumab in subjects with glioblastoma. Patients were enrolled into 5 non-comparative cohorts receiving either durvalumab monotherapy or durvalumab and bevacizumab combotherapy. MOS was 15.1 months (95% CI 12.0–18.4).	159	[22]
NCT01454596	Phase I/II SA	A study to determine the safety and effects of CART-EGFRvIII therapy in patients with recurrent GBM. CAR T-cell therapy was given in combination with a synthetic IL-2 molecule, aldesleukin, and a lymphodepleting preparative regimen of cyclophosphamide and fludarabine. MOS was 6.9 months (IQR 2.8–10.0), with 3 instances of adverse effects.	18	[23]
NCT02454634	Phase I SGA	A study to identify the safety and tolerability of the first in-human mutant IDH1 peptide vaccine in patients with WHO Grade III–IV gliomas. Vaccine-induced immune responses were observed in 93.3% of patients. No regime-limiting toxicity was observed.	32	[24]
NCT01250470	Phase I SGA	A study of the side effects of a vaccine therapy directed against the tumorigenic molecule survivin, in combination with the synthetic granulocyte-macrophage colony-stimulating factor sargramostim. The therapy was well tolerated with no serious adverse events attributable to the therapy; 6 out of 8 immunologically evaluable patients developed immune responses to the vaccine.	9	[25]
NCT05163080	Phase II R, PA, DB	A Phase II clinical trial analyzing whether an antisurvivin vaccine treatment combined with SOC TMZ treatment is better than TMZ treatment alone for GBM patients. The primary outcome is OS. The trial is in progress.	265	[26]
NCT00157703	Phase I SGA	A study to determine the safety of oncolytic HSV-1, G207, given in combination with radiation for recurrent GBM. Three serious AEs were reported (seizures after administration), possibly related to G207 administration. The estimated median survival time from G207 inoculation was 7.5 months (95% CI 3.0–12.7).	9	[27]
NCT02457845	Phase I SGA	A study to determine the safety of G207 treatment in combination with radiotherapy for pediatric patients with recurrent supratentorial brain tumors. Twenty Grade 1 AEs were reported, possibly related to G207. MOS was 12.2 months (95% CI 8.0–16.4) as of 6/2020. The trial is in progress.	12	[28]
NCT00589875	Phase II SGA	A study to determine the safety and potential efficacy of adenoviral vector expressing HSV1-tk (aglatimagene besadonevac, AdV-tk) followed by valacyclovir in combination with the SOC treatment. MOS was 17.1 month for treatment + SOC vs. 13.5 months for SOC alone (*p* = 0.0417)	52	[29]
EudraCT 2004-000464-28	Phase III R, PA	A study comparing the adenovirus vector-mediated delivery of HSV1-tk (AdV-tk) followed by IV ganciclovir with the SOC treatment versus SOC treatment alone in newly diagnosed GBM. No difference in MOS was found in the experimental (497 days, 95% CI 369–574) versus the control group (452 days, 95% CI 437–558) (HR 1.18, 95% CI 0.86–1.61, *p* = 0.31).	250	[30]
NCT01491893	Phase I SA	A study to determine the maximum tolerated dose of a live attenuated polio–rhinovirus chimera (PVSRIPO) on GBM; 19% of patients treated with PVSRIPO had a Grade 3 or higher adverse event.	61	[31]
NCT02414165	Phase II/III R, PA	A study of a gamma retroviral replicating vector encoding a yeast cytosine deaminase, vocimagene amiretrorepvec, combined with 5-fluorocytosine treatment versus SOC in recurrent GBM. MOS was 11.10 months for the experimental group compared to 12.22 months for the control group (HR = 1.06; 95% CI 0.83, 1.35; *p* = 0.62).	58	[32,33]

DB, double-blind; PA, parallel assignment; R, randomized; SA, sequential assignment; SGA, single group assignment.

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
