# Peer review of "Advances in Immunotherapy for the Treatment of Adult Glioblastoma: Overcoming Chemical and Physical Barriers"

_cancers, 2022, doi:10.3390/cancers14071627_

Round 1
Reviewer 1 Report
In the present review, authors discusses the immunotherapy as potential treatment strategy for glioblastoma and challenges that need to be managed. I have several reservations, my comments are appended as below:
- Abstract need not to be annotated with reference.
- Line 59-63 annotate reference on ultrasound.
- While introducing immunotherapy, authors should first discuss its components as inhibitory receptors other than PD-L1, CTLA4 and other immune cells.
- Information on trials: line 94- this seems valuable information. Authors should add a table and list approved as well as undergoing trials. This should include trial no, reference, details of trial in brief and patient no. Same for other modules discussed.
- While discussing the treatment module, authors should discuss shortcomings in separate heading. I observe that authors do note for immunotherapy, similarly should be discussed for OVT and vaccines.
- Each section should end with summary in brief and possible future directions.
- While discussing studies in patients, authors should note the number and statistical inference (HR, P value).
- Authors should elaborate the role of stem cells in treatment resistance.
- For immunotherapy, other cofounders as BMI, smoking are known to play an important role. Authors may refer PMID: 33076303and add a para.
- Mutation burden- discuss in few lines how TMB is determined.
- Line 436-437- specify the interleukins and immune cells.
- Strategies to enhance immunotherapy effectiveness: authors should also discuss limitations with spate heading.
- There should be future directions section.
Author Response
Please see attachment for reviewer responses and note that line numbers refer to document with changes tracked.

Reviewer 2 Report
It was a review paper about the different approaches used for overcoming the barriers towards the treatment of glioblastoma via immunotherapy. Here are some comments on this study that should be considered before publication:
- Please introduce all the abbreviations at the first time usage.
- Please rewrite this sentence "This is notable as it may augment the use of MRI in disease surveillance as new therapies are further evaluated."
- "The increased permeability has been observed in normal brains, brains affected by neurodegenerative diseases (e.g. Parkinson’s Disease and Alzheimer’s Disease), and brain tumors." Are you sure? Why normal brains show increased permeability?
- Please check all the references again. Some of them have different format. Moreover, some of the references are out of date please updating them.
- Please also add some eye-catching figures used in experimental researches.
- Describe about the targeted immunotherapy via nanoparticles in section 4.
Author Response

(The authors gave the same response as above.)

Round 2
Reviewer 1 Report
All my concerns are adressed.
Reviewer 2 Report
Thanks for addressing the comments.